# Unpacking In-Context Learning: Underlying Mechanism and Out-of-Distribution Generalization via Blended Training on Function Mixture

## Abstract

Transformer-based language models have achieved remarkable success across a wide range of real-world tasks, yet the internal mechanisms that govern their behavior remain only partially understood. Recent research has increasingly focused on the phenomenon of in-context learning (ICL) and its ability to generalize beyond the training distribution. However, many of these studies are conducted under simplified conditions, where both training and evaluation use prompts derived from a single, clearly defined function. As a result, it remains unclear how models behave in more structurally diverse or ambiguous settings. In this study, we examine ICL under a blended training paradigm, in which each training prompt contains examples sampled from multiple function classes, without any explicit task identifiers or structural signals. Using standard ICL benchmarks such as linear and quadratic classification, we assess how this training approach influences model behavior, robustness, and generalization. Our findings indicate that under blended training, the commonly observed function selection behavior, where the model implicitly identifies and applies a single underlying function, plays a less central role. Instead, the model demonstrates more flexible pattern recognition, improved resilience to input noise, and stronger generalization to out-of-distribution tasks. These results suggest that training on structurally mixed prompts can enhance a model's adaptability in unfamiliar scenarios.

## 1 Introduction

In-context learning (ICL) is among the most intriguing and under-explored capabilities of large transformer-based language models (LLMs). By conditioning on input-output examples provided in a prompt, models can perform novel tasks without any parameter updates—a behavior popularized by GPT-3 (Brown et al., 2020). This surprising ability has sparked growing interest in understanding how such behavior emerges, and whether it reflects pattern matching, implicit optimization, or generalization across tasks.

A prominent perspective frames ICL as a form of *function learning* (Garg et al., 2022): given a sequence of input-output pairs, the model is expected to infer the latent function that generated them and generalize to unseen inputs. Under this view, the prompt acts as a support set from which the model recovers the governing rule, and the prediction for the next query point reflects the model's inductive biases. This interpretation has guided much of the theoretical and empirical work on ICL (Zhang et al., 2023; Huang et al., 2023; Cheng et al., 2024; Guo et al., 2024; Yang et al., 2024). Such a framework holds strong practical potential, particularly in domains like stock market forecasting, where underlying patterns are often non-stationary and span a variety of dynamics. A model capable of flexible adaptation across task types, without requiring parameter updates or retraining, offers a scalable and efficient alternative for real-world decision-making.

On the other hand, Hollmann et al. (2022); Müller et al. (2022) proposed Prior-Data Fitted Networks (PFNs), revealing that transformer-based models can approximate Bayesian inference when trained offline on carefully constructed task distributions. These works suggest that ICL involves structured,

algorithmic behavior within the model. Subsequent analyses have examined how this behavior scales to more diverse or challenging task distributions.

Several works highlight structured, phase-wise computation in transformers, showing how attention heads specialize in roles such as preprocessing, optimization, or extrapolation (Anonymous, 2024; Bhasin et al., 2024; Chen et al., 2024). At the same time, questions remain about the generalization capabilities of ICL, particularly in out-of-distribution (OOD) settings. Some studies, using different methodologies, have pointed out that models possess generalization capabilities (Song et al., 2025; Wang et al., 2025; Bhasin et al., 2024). However, high task diversity may weaken Bayesian-like behavior (Raventos et al., 2023). Even with multi-task pretraining, generalization tends to deteriorate when tasks are highly ambiguous or misaligned with the model's inductive biases (Yadlowsky et al., 2023; Bhasin et al., 2024). These insights raise important questions about how generalization arises in ICL, and whether it necessarily depends on task abstraction, model capacity, or training distribution.

Li et al. (2024b) introduced *blended training*, in which each prompt contains examples sampled from multiple function families without explicit structural cues. However, their results reported only the performance, the implications of blended training for model behavior and generalization remain underexplored. Building on this line of inquiry, in this work, we revisit and extend blended training in the context of ICL, systematically evaluating its performance on self-designed function sets and probing its mechanisms. Our contributions are as follows:

- **Performance validation:** We build upon the blended training paradigm (Li et al., 2024b) and confirm that it achieves comparable accuracy to vanilla training methods on common ICL benchmarks involving multiple function classes.

- **Mechanism analysis:** We empirically analyze the internal mechanisms of vanilla-trained and blended-trained models through controlled experiments and demonstrate that their behavior challenges commonly held assumptions about function selection and lowest-error preference.

- **Robustness and generalization:** We show that blended training enhances both noise robustness and out-of-distribution generalization, even in cases where recent literature suggests ICL tends to overfit to the training distribution.

## 2 RELATED WORK

### 2.1 IN-CONTEXT LEARNING AS FUNCTION LEARNING

In-Context Learning (ICL) describes the surprising ability of large language models (LLMs) to perform tasks without parameter updates, by conditioning solely on a prompt containing example input-output pairs. In this setup, a model is given a context consisting of $(x_1, y_1), (x_2, y_2), ..., (x_{k-1}, y_{k-1})$ and is asked to predict the output $y_k$ corresponding to a new input $x_k$. No task identifier or training labels are provided at inference time, the model must infer the latent function $f$ governing the relationship between inputs and outputs in-context.

A pioneering view of ICL as function learning was introduced by Garg et al. (2022), who demonstrated that transformers trained on synthetic supervised tasks could recover various function classes purely through context. The model is viewed as approximating a function $f$ from a hypothesis space, with the prompt acting as a support set for generalization. Subsequent work has expanded the space of tested functions with deeper behavior analysis, including linear functions (Raventos et al., 2023; Wu et al., 2024; Akyürek et al., 2023), boolean functions (Bhattamishra et al., 2024), dynamical systems (Li et al., 2023b) and even neural networks (Wang et al., 2024b). These benchmarks serve as a foundation for probing ICL's inductive behavior, and provide controlled settings to evaluate mechanism, performance, and generalization.

### 2.2 MULTI-FUNCTION CONTEXTS AND TASK MIXTURE

While early studies in ICL typically focused on prompts where all input-output pairs came from a single underlying function (Garg et al., 2022; Wu et al., 2024; Bhattamishra et al., 2024; Li et al., 2023b; Wang et al., 2024b), recent work has explored broader task distributions involving mixtures

of functions and increased task ambiguity. Some recent studies have introduced functional diversity during training, sampling from multiple function classes to analyze generalization across tasks or uncover underlying mechanisms (Yadlowsky et al., 2023; Wang et al., 2024a; Tripuraneni et al., 2024; Li et al., 2023a). However, their evaluation protocols typically remain structured, with each prompt at test time derived from a single function class. This preserves a consistent functional identity within the context and implicitly encourages models to specialize or recover that function.

Blended training, introduced by Li et al. (2024b), presents a less structured setting where each training prompt consists of examples from multiple function classes, without task identifiers, segmentation tokens, or ordering structure. The model receives a mixed context of input-output pairs $(x_1, y_1), (x_2, y_2), \ldots, (x_{k-1}, y_{k-1})$, with each $y_i$ generated by a function $f_j$ sampled from a set of functions. This setup allows us to investigate how models handle ambiguity and whether they rely on global function identity or adapt to local patterns. In this work, we use blended training to explore the emergence of attention structures, robustness to noise, and generalization to out-of-distribution tasks.

### 2.3 ATTENTION ANALYSIS IN ICL

Attention-based interpretability has played a central role in uncovering in-context learning (ICL) mechanisms. Several specialized attention patterns have been identified across layers and heads. One important discovery is the *induction head* (Anonymous, 2024; Song et al., 2025), a head that copies tokens forward in the prompt by focusing attention on the previous token position. Induction heads are thought to facilitate extrapolation and enable sequence pattern matching. More recently, *retrospective heads* have been proposed (Bhasin et al., 2024). These heads attend backward across the prompt to identify examples similar to the current query input, acting as a kind of in-context nearest-neighbor retriever. Retrospective attention appears to play a critical role in model behavior when tasks are ambiguous or not easily classifiable by position.

In our work, we assess the importance of each attention head using a masking-based diagnostic adapted from Chen et al. (2024). For each attention head $j$ in layer $i$, we zero out its output and compute the resulting drop in prediction accuracy:

$$\Delta\text{Acc}^{(i,j)} = \text{Acc}_{\text{full}} - \text{Acc}_{\text{masked}(i,j)}$$

where $\text{Acc}_{\text{full}}$ is the original model accuracy, and $\text{Acc}_{\text{masked}(i,j)}$ is the accuracy when that specific head is ablated. We then normalize each head's impact within its layer as:

$$W_{i,j} = \frac{\Delta\text{Acc}^{(i,j)}}{\sum_k \Delta\text{Acc}^{(i,k)}}$$

resulting in a heatmap that quantifies the relative importance of all heads in each layer.

This diagnostic approach complements structural analysis by quantifying the functional role of each head in ICL behavior, offering insight into how different attention pathways contribute to prompt interpretation and prediction under various training regimes.

### 2.4 GENERALIZATION TO OUT-OF-DISTRIBUTION FUNCTIONS

The structure of this section follows prior work (Wang et al., 2025), which outlines several perspectives on how in-context learning may generalize to out-of-distribution settings. The extent to which ICL generalizes beyond its training distribution remains a topic of active debate. Several theoretical frameworks have been proposed to explain generalization behavior:

- **Bayesian inference.** The model infers a latent task concept from the prompt and makes predictions accordingly (Xie et al., 2022; Wies et al., 2023; Müller et al., 2022). However, these frameworks often leave the process of task inference implicit, especially in OOD settings.

- **Gradient descent emulation.** Transformers may internally perform gradient-based optimization. Prior works (Shen et al., 2024; von Oswald et al., 2023) construct architectures where ICL mimics a linear regression solver trained by gradient descent.

- **Function or algorithm selection.** The model chooses a function from a set of pre-trained routines (Bai et al., 2023; Wang et al., 2024a). This view suggests brittle generalization when encountering functions outside the training distribution.
- **Retrieval-based reasoning.** Some studies argue that the model retrieves in-context examples with similar inputs to the query using attention Li et al. (2024a).

We focus on the function selection hypothesis here, which has been supported by several theoretical and empirical studies (Wang et al., 2025), arguing that models tend to select functions that minimize test-time error. We ask whether models trained under either vanilla or blended regime must select from learned functions, or if they can flexibly fit to in-context patterns with a more general "super function".

## 3 TASK DESIGN AND GENERALIZATION SETTINGS

To investigate the mechanisms and generalization behavior of in-context learning under different training paradigms, we design controlled synthetic tasks where the target functions are explicitly defined. This enables a precise evaluation of model behavior and internal representations.

Table 1: Function types and their corresponding inequality-based decision rules used in different test settings.

| Function name | Category | Inequality |
|---|---|---|
| Linear classification (LC) | (1), (2) | $f_{LC}(x) = 1[w^\top x > 0]$ |
| Checkerboard classification (CC) | (1) | $f_{CC}(x) = 1[(w_1^\top x)(w_2^\top x) > 0]$ |
| Quadratic classification (QC) | (2) | $f_{QC}(x) = 1[x^\top A x > 0]$ |
| Residual classification (R) | (2), (3) | $f_R(x) = 1[x_j > \tau], \; j \in \{1, \ldots, d\}$ |
| General quadratic classification | (3) | $f(x) = 1[x^\top A x + w^\top x + b > 0]$ |

### 3.1 CATEGORY (1): LC VS. CC BINARY TASK MIXTURE

In the first category, we construct a binary function mixture consisting of two qualitatively distinct classification tasks: Linear classification (LC) and Checkerboard classification (CC).

The design of the LC and CC task ensures that their decision boundaries are fundamentally misaligned, making it impossible for a classifier trained on one to correctly predict the other. Specifically, attempting to solve the Checkerboard Classification (CC) task using a model optimized for Linear Classification (LC) results in near-random performance ($\sim 50\%$ accuracy), and vice versa. (Detail in Appendix: see Fig 3)

### 3.2 CATEGORY (2): QC VS. LC VS. R MULTIPLE TASK MIXTURE

In this category, we aim to test whether the model can simultaneously handle three different functions, particularly under the blended training setting. Furthermore, combining these three function classes can provide better insights in the analyses of generalization (see Category 3) and mechanism, and allows us to evaluate the model's limits when faced with more complex or ambiguous contexts. Specifically, we train the model using three function types: Quadratic classification (QC), Linear classification (LC) and Residual classification (R).

### 3.3 CATEGORY (3): FUNCTIONS USED TO TEST GENERALIZATION

This category tests the model's ability to transfer patterns from seen tasks to more complex, unseen contexts, evaluating generalization. We introduce held-out test functions that differ structurally from the training tasks to assess out-of-distribution (OOD) generalization.

For category 1 (LC and CC), we use Residual Classification (R) as the OOD function, since R introduces a distinct decision structure with axis-aligned thresholding, unlike the spatial regularities of CC and QC.

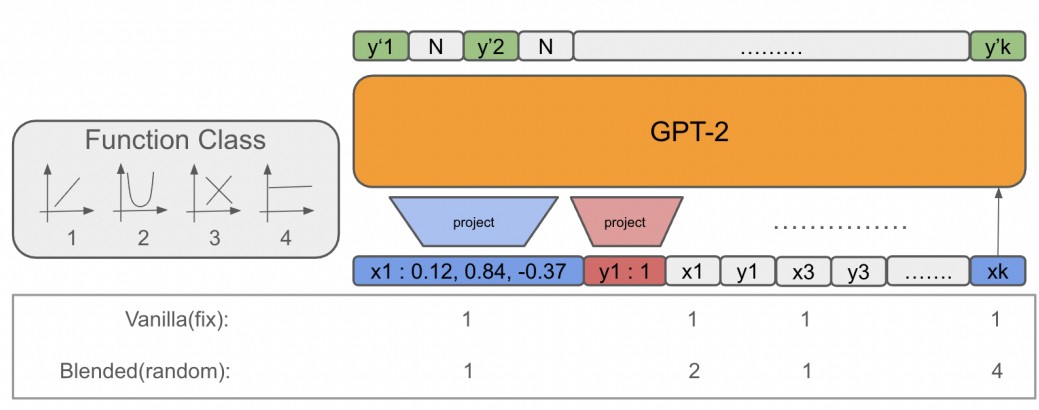

Figure 1: Illustration of the training diagram: GPT-2 processes input-output pairs drawn from either a single (vanilla) or a mixed (blended) set of function classes.

For category 2 (LC, QC, and R), we use a general QC function with unseen parameterizations, providing a shifted but structurally related function class. This challenges the model to apply abstractions learned from multiple tasks in novel configurations.

## 4 TRAINING AND EVALUATION SETUP

### 4.1 TRAINING DETAIL

During training, each input sequence consists of 100 input-output pairs. The model observes 99 context points $(x_1, y_1), (x_2, y_2), ..., (x_{99}, y_{99})$ and is trained to predict the target output $y_{100}$ corresponding to the input $x_{100}$. Both the inputs x and the underlying function weights w are independently sampled from a standard normal distribution $\mathcal{N}(0, 1)$. The model architecture is based on GPT-2, which processes the sequence of tokenized input-output pairs using separate embedding layers for x and y before feeding them into the transformer layers.

Two training strategies are compared: vanilla and blended. In the vanilla setting, all points within a training example are sampled from the same function class (e.g., linear, quadratic), and this function remains fixed throughout the prompt. In contrast, the blended setting introduces greater variability by randomly selecting a function class for each point in the context, with each function class chosen uniformly from a predefined set. For each training instance, the model predicts the final output $y_{100}$ based on the full context of 99 preceding points. The predicted values $\hat{y}_1, \hat{y}_2, ..., \hat{y}_{100}$ are compared with the ground truth targets to compute the loss, and the performance of the model is tracked over time for both training settings.

### 4.2 EVALUATION METHOD

To evaluate the performance of the model, the following method was used. In each trial, a context of 99 points was randomly sampled, and the 100-th point was appended 2000 times to assess prediction accuracy within that context. This procedure was repeated 1000 times, and the average accuracy across all trials was reported as the final result.

## 5 EXPERIMENTAL RESULT

### 5.1 PERFORMANCE VALIDATION

To evaluate whether blended training impacts in-context learning performance, we compare models trained under blended and vanilla supervision across several function classes, including linear classification (LC), quadratic classification (QC), residual classification (R), and checkerboard classification (CC). As shown in Table 2 and Table 3, the blended-trained model achieves accuracy

Table 2: Accuracy (%) for LC and CC binary task mixture under different training modes.

| Training Mode | LC (%) | CC (%) |
|---|---|---|
| blended | 98.60 | **95.90** |
| vanilla | **98.80** | 95.65 |

Table 3: Accuracy (%) for LC, QC, and R multiple task mixture under different training modes.

| Training Mode | LC (%) | QC (%) | R (%) |
|---|---|---|---|
| blended | **98.70** | 95.50 | 98.15 |
| vanilla | 98.45 | **96.60** | **99.50** |

comparable to, and in some cases slightly surpasses, that of the vanilla-trained model. These results suggest that mixing function classes during training does not degrade predictive performance. It is worth noting that for evaluation, both vanilla- and blended-trained models are tested using vanilla-style contexts, where all input-output pairs within a sequence are generated from the same underlying function. This ensures a fair comparison of how well each model generalizes to individual function classes.

## 5.2 MECHANISM ANALYSIS

To further investigate whether the function selection hypothesis holds under different training regimes, we conduct a series of experiments, each accompanied by two explicitly formulated hypotheses. These experiments are designed to probe the underlying mechanisms behind model behavior and examine whether the model truly performs function selection or instead adapts dynamically based on the presented context.

### 5.2.1 (1) OUT-OF-DISTRIBUTION FUNCTION TEST

Table 4: Accuracy of different models tested with binary task mixture (setting 1) and multiple task mixture (setting 2).

|  | vanilla | blended | LC | CC/QC | R | mix |
|---|---|---|---|---|---|---|
| **setting 1** | 0.8495 | **0.8905** | 0.7381 | 0.6985 | – | 0.8214 |
| **setting 2** | 0.8312 | **0.8637** | 0.6265 | 0.7909 | 0.6657 | 0.8144 |

In this experiment, we evaluate whether models can generalize to out-of-distribution (OOD) function types that were not seen during training. Specifically, we compare vanilla-trained and blended-trained models to a baseline formed by individually trained models, that is, models trained on a single function type without exposure to other task types. Two evaluation settings are considered:

- **Setting 1:** The model is trained on Category 1 (LC and CC), and tested on the residual classification (R) task.

- **Setting 2:** The model is trained on Category 2 (LC, QC and R), and tested on a general quadratic classification task composed of mixed distributions (LC + QC + R).

To benchmark performance, we introduce a **Mix baseline**, defined as the maximum accuracy across individually trained models in the same round (i.e., $\max(\text{LC}, \text{QC}, \text{R})$ or $\max(\text{LC}, \text{CC})$). If a model merely memorizes function-specific routines, its performance should not exceed the Mix baseline. We consider two hypotheses:

- **H1:** The model internalizes specific function classes during training. Consequently, its generalization should be no better than the best of the singly trained models.

- **H2:** The model does not encode functions internally but instead adapts to contextual information. This allows it to go beyond the known function classes and generalize to new combinations.

As shown in Table 4, both the vanilla and blended models consistently achieve higher accuracy than the Mix baseline in both settings. This performance gap suggests that the models are not merely selecting among pre-learned functions, but are instead adapting based on the presented context. These results support H2 and call into question the function selection hypothesis.

### 5.2.2 (2) MODEL BIAS TEST

Table 5: Comparison under input point replacement (x — y indicates number of LC — CC being selected out of 100 attempts).

| model | replace 0 pts | replace 2 pts | replace 5 pts | replace 10 pts |
|---|---|---|---|---|
| blended | 99 — 1 | 89 — 11 | 57 — 43 | 2 — 98 |
| vanilla | 100 — 0 | 59 — 41 | 19 — 81 | 4 — 96 |

This experiment investigates whether the model exhibits bias when interpreting ambiguous prompts. We construct contexts that resemble both linear classification (LC) and checkerboard classification (CC), then incrementally replace a small number of points to lean toward CC. We test 100 such prompts per setting, comparing the model's classification accuracy on LC and CC. Whichever function achieves higher accuracy is considered the one "selected" by the model. We evaluate two hypotheses:

- **H1**: The model selects the function that minimizes expected error.
- **H2**: The model does not explicitly evaluate error but relies on internal biases or heuristics.

As shown in Table 5, both models initially prefer LC (Blended: 99/100; Vanilla: 100/100). As more points are replaced toward CC, the vanilla model shifts its preference earlier (e.g., 81 CC selections at 5-point replacement), while the blended model remains more committed to LC (57 LC vs. 43 CC). This suggests that the blended model may exhibit a stronger bias toward LC.

These findings contradict the lowest-error selection hypothesis (H1). If the model were simulating error, it should be indifferent under ambiguity and shift decisively once the evidence favors one function. Instead, the model shows a preference-driven response, supporting H2. While this does not conclusively rule out broader function selection, it highlights that "lowest-error selection" is not the strategy the model follows in ambiguous contexts.

### 5.2.3 (3) ATTENTION HEAD ANALYSIS

This experiment investigates whether certain attention heads specialize in solving specific tasks, namely, linear classification (LC) or checkerboard classification (CC). To evaluate this, we adopt an ablation-based approach: systematically zeroing out each attention head and measuring the corresponding drop in model accuracy.

For each model, we generate an accuracy difference heatmap, where each cell represents the drop in accuracy after ablating a specific head. In each heatmap, rows correspond to layers and columns to head indices. Larger values indicate heads that are more critical to model performance. We consider two competing hypotheses:

- **H1**: The model encodes abstract functions. Specific heads contribute exclusively to Function A or B, allowing the model to select the appropriate function based on the prompt.
- **H2**: The model does not explicitly encode function identity. Instead, attention heads serve as general-purpose mechanisms that support multiple functions simultaneously.

As shown in the heatmaps (Fig 2), ablating top-performing heads leads to accuracy drops across both LC and CC tasks. Notably, those influential heads tend to overlap between tasks, suggesting

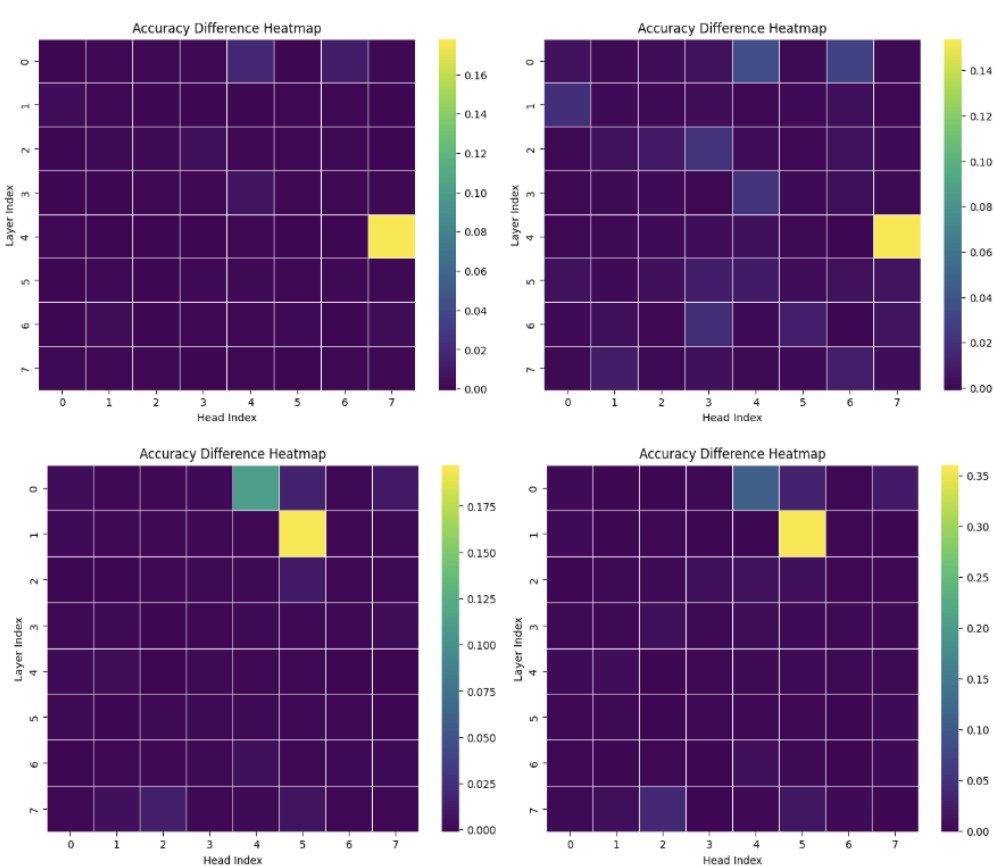

Figure 2: Heatmaps of accuracy drop in models(Top: vanilla, Bottom: blended)

that the attention heads are not function-specific modules, but rather shared components used for flexible contextual fitting.

This support **H2**: attention heads contribute to flexible pattern fitting, rather than acting as modular components dedicated to specific functions. The shared influence of top heads across LC and CC undermines the function-selection hypothesis and suggests that the model processes prompts holistically: dynamically adapting its computation based on input structure, rather than invoking fixed routines aligned to specific function classes.

### 5.3 GENERALIZATION AND ROBUSTNESS

To assess the effectiveness of different training strategies beyond accuracy alone, we further evaluate the models' generalization and robustness. Specifically, we design experiments that separately target these two aspects: the ability to extrapolate to unseen functional compositions, and the resilience to noisy input conditions. By comparing blended, vanilla, and noise-augmented models under controlled settings, we aim to determine whether the benefits of blended training extend meaningfully into these challenging scenarios.

#### 5.3.1 OOD GENERALIZATION COMPARISON WITH NOISE-AUGMENTED MODEL

We further assess the advantage of blended training by introducing a noise-augmented baseline. The setup follows that of (2) Out-Of-Distribution Function Test in Section 5.2.1, with the addition of a model trained on noisy contexts, where a random subset of values is flipped (0 to 1 or vice versa) with probability 0.3. This noise-augmented model serves as a control, testing whether blended training merely benefits from noise-based regularization.

Table 6: Generalization accuracy across different models.

|  | vanilla | blended | noise |
|---|---|---|---|
| setting 1 | 0.8551 | **0.8960** | 0.8863 |
| setting 2 | 0.8312 | **0.8620** | 0.8270 |

As shown in Table 6, the blended model outperforms both the vanilla and noise-augmented models (noted as **Noise**) across over half of the 1000 contexts. In Setting 1, the blended model achieves 0.8960, surpassing the vanilla (0.8551) and noise-augmented (0.8863) models. In Setting 2, the blended model (0.8620) consistently outperforms both the vanilla (0.8312) and noise-augmented (0.8270) models, underscoring its robustness. These results demonstrate that the benefits of blended training are not solely due to noise, as the noise-augmented model struggles in more complex settings.

### 5.3.2 ROBUSTNESS UNDER NOISY INFERENCE

Table 7: Accuracy under different noise levels (0.1, 0.2, 0.3) for each task, comparing Vanilla-, Blended-, and Noise-augmented models.

| Setting | Task | Noise Level = 0.1 | | | Noise Level = 0.2 | | | Noise Level = 0.3 | | |
|---|---|---|---|---|---|---|---|---|---|---|
|  |  | Vanilla | Blended | Noise | Vanilla | Blended | Noise | Vanilla | Blended | Noise |
| Setting 1 | $LC'$ | 0.81 | **0.98** | 0.97 | 0.53 | 0.92 | **0.93** | 0.50 | **0.68** | 0.64 |
|  | $CC'$ | 0.85 | **0.93** | **0.93** | 0.74 | 0.83 | **0.86** | 0.61 | 0.65 | **0.69** |
| Setting 2 | $QC'$ | 0.88 | **0.94** | **0.94** | 0.77 | 0.87 | **0.89** | 0.61 | 0.66 | **0.69** |
|  | $LC'$ | 0.89 | **0.97** | **0.97** | 0.63 | **0.93** | **0.93** | 0.53 | **0.73** | 0.70 |
|  | $R'$ | 0.87 | **0.98** | **0.98** | 0.70 | 0.94 | **0.96** | 0.58 | **0.76** | 0.72 |

This experiment evaluates the robustness of different training strategies under varying levels of inference-time noise. We compare vanilla, blended, and noise-augmented models across five tasks in two settings, introducing random flip during inference at noise levels of 0.1, 0.2, and 0.3. Each cell in the accuracy table reports performance at these three levels, respectively. We consider two training configurations:

- **Setting 1**: Train on LC and CC → Test on $LC'$ and $CC'$ under noise
- **Setting 2**: Train on LC, QC, and R → Test on $LC'$, $QC'$, and $R'$ under noise

As shown in Table 7, both blended and noise-augmented models outperform the vanilla baseline across all noise levels and tasks. Notably, the blended model matches or exceeds the noise-augmented model's robustness, achieving 0.97, 0.93, and 0.73 in the "$LC'$ (Setting 2)" setting, compared to 0.97, 0.93, and 0.70 for the noise-augmented model. These results suggest that blended training, by exposing the model to diverse functional patterns, inherently enhances robustness to input noise, promoting stable decision boundaries that generalize well even under noisy conditions.

## 6 CONCLUSION

In this work, we investigated the effects of blended training and its implications for in-context learning mechanisms. Our results demonstrate that blended training achieves comparable accuracy to vanilla training, suggesting that incorporating functional diversity does not compromise predictive performance. We further examined the function selection hypothesis through four targeted experiments. In all the experiments, both blended and vanilla models exhibited behaviors inconsistent with the hypothesis. These findings indicate that function selection may not adequately explain model behavior under different training strategies. Finally, our comparison with models trained under random noise reveals that blended training offers stronger generalization and similar robustness to noisy inputs, despite not relying on explicit noise injection. This suggests that blended training provides benefits beyond noise-based regularization, promoting more stable and adaptive inference in ambiguous or degraded conditions.

ACKNOWLEDGEMENTS

We would like to acknowledge the assistance of OpenAI's ChatGPT (GPT-5) in refining and condensing parts of this manuscript. The model's support in improving clarity and conciseness has been invaluable in enhancing the overall quality of this work. We express our sincere gratitude for this helpful tool.

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

## A    TRAINING CONFIGURATION

Table 8: Training configuration and data generation setup.

| parameter | value |
| --- | --- |
| number of points ($k$) | 100 (99 context + 1 evaluation) |
| data size | $64 \times 300{,}000$ (batch size $\times$ epochs) |
| input dimension (dim) | 3 |
| learning rate (lr) | 0.001 |
| input distribution ($x$) | $\mathcal{N}(0, 1)$ |
| weight distribution ($w$) | $\mathcal{N}(0, 1)$ |
| gpu | NVIDIA GeForce RTX 3090 |

The model was trained with a batch size of 64 in 300000 epochs, so the total number of data sequences is $64 \times 300000$. The latent dimension $k$ is configured to 3 to facilitate both prompter training and visualization. Training was conducted using a learning rate of 0.001. All experiments were performed on an Nvidia GeForce RTX 3090 GPU.

## B    REASONS FOR TASK DESIGN: CATEGORY (1)

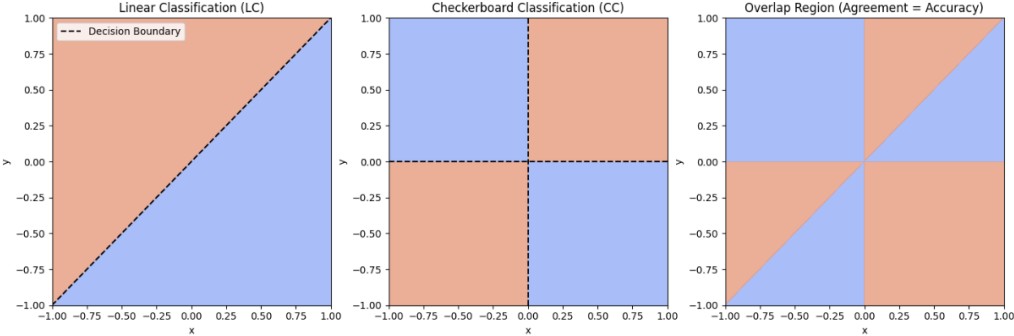

Figure 3: Visual illustration of task incompatibility. Left: linear classification (LC) with a single decision boundary. Middle: checkerboard classification (CC) with alternating labels based on sign agreement. Right: overlap region showing prediction agreement between LC and CC.

The 50% guarantee stems from the symmetry and structure of the CC function: it labels points as class 1 when the signs of two linear projections agree (both positive or both negative), and class 0 otherwise. Consequently, CC has alternating positive and negative regions that cannot be separated by a single hyperplane. On the other hand, LC is based on a single hyperplane decision boundary, which inherently fails to capture the XOR-like structure embedded in CC. This incompatibility provides a natural lower bound: any attempt to apply an LC decision rule to CC-labeled data (or vice versa) will produce accuracy close to 50%, reinforcing that high accuracy on both must come from task-specific generalization rather than naive function selection or averaging.

## C    ADDITIONAL RESULTS

### C.1    VISUALIZATION OF MODEL CLASSIFICATION RESULTS

To better understand the model's behavior, we visualize the prediction pattern from the blended-trained model given an input prompt in Figure 4. Specifically, we construct a context of 99 input-output pairs and sequentially test the 100-th query point. Each predicted point is then plotted in 3D space, colored according to the model's output: blue for class 1 and red for class 0. As shown in the figure, the model's predictions reflect clear and structured separation, suggesting that it successfully

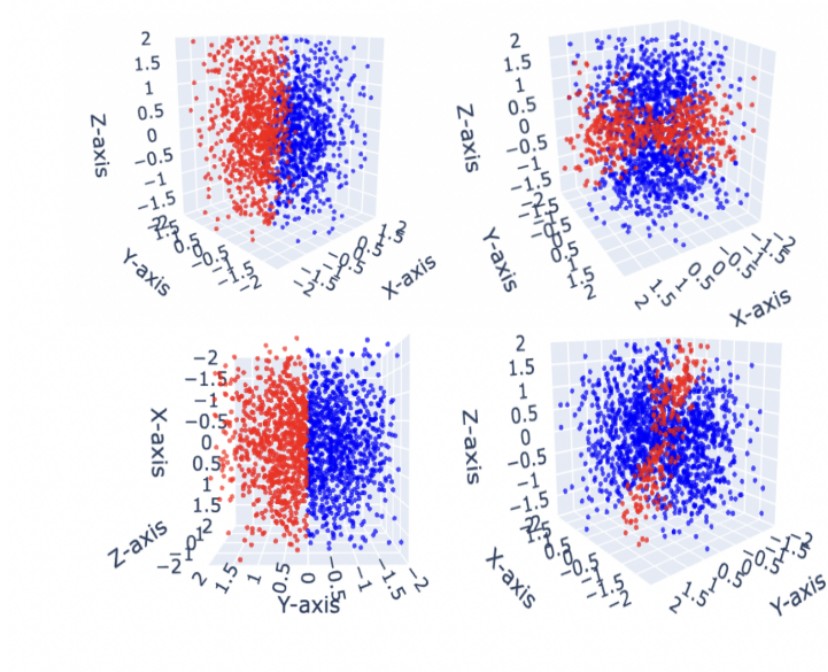

Figure 4: Visualization of 3D classification data from four different functions. Top-Left: LC, Top-Right: QC, Bottom-Left: R, Bottom-Right: CC

extrapolates the correct decision boundary from the observed examples. In particular, the predicted decision structure aligns well with the underlying task, indicating that the model has internalized not just individual examples but the generative rule behind them.

## C.2 ADDITIONAL TEST FOR MECHANISM ANALYSIS: FUNCTION MIXTURE TEST

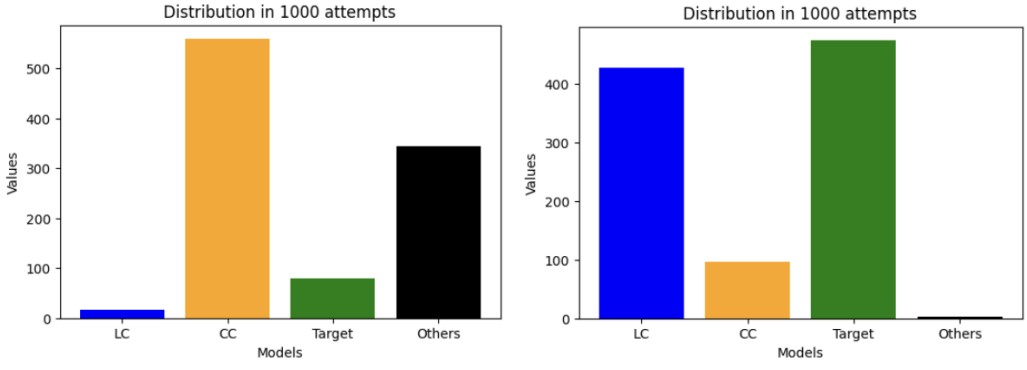

Figure 5: Distribution of model's choices over 1000 attempts. Left: Vanilla, Right: Blended

In this experiment, we examine the model's behavior under the binary task mixture setup (Category 1), where linear classification (denoted as Function A) and checkerboard classification (Function B) are presented within the same prompt. As mentioned in Section 3.1, these two tasks are structurally incompatible: attempting to solve one using the decision boundary of the other yields accuracy close to random guessing (approximately 50%). This property allows us to probe whether the model is rigidly selecting between known functions or flexibly adapting to contextual cues. We then formulate two hypotheses:

- **H1**: The model has memorized how to solve either A or B and, when given a mixed context, simply falls back on one of the known routines.

- **H2**: Although trained to solve A and B, the model does not explicitly internalize the functions themselves. Instead, it learns how to fit examples based on contextual patterns, preserving flexibility to adapt to novel or ambiguous mixtures.

To test these hypotheses, we construct prompts containing a mixture of A and B examples (99 points), and evaluate the model's prediction on a 100th point sampled from either function. If the model achieves accuracy above 60% on both A and B tasks within the same prompt, this suggests that it is not merely selecting between known routines, but is capable of interpolating behavior based on the input distribution. We evaluate this behavior over 2000 samples and categorize each prompt into one of four groups based on model performance:

- **LC (blue)**: The model shows strong accuracy on the linear task ($acc_{lc} > acc_{cc} + 0.2$ and $acc_{cc} < 0.6$), indicating preference for solving A.

- **CC (orange)**: The model favors the checkerboard task ($acc_{cc} > acc_{lc} + 0.2$ and $acc_{lc} < 0.6$), indicating preference for solving B.

- **Target (green)**: The model achieves non-trivial performance on both functions ($acc_{lc} > 0.6$ and $acc_{cc} > 0.6$), suggesting successful adaptation to both tasks.

- **Others (black)**: Prompts that do not meet any of the above conditions, including borderline or ambiguous cases.

As shown in the bar plots (see Figure 5), in the blended training condition, over 400 of the tested prompts fall into the **Target** category. This outcome supports H2, indicating that the model does not rigidly select a single function but instead adapts dynamically based on the prompt, even in the absence of explicit function labels.

C.3    MORE DETAIL ON ATTENTION HEAD ANALYSIS

Table 9: Overlap ratios of top-$k$ influential attention heads across tasks.

| method | layer | head | ratio (top-5) | ratio (top-10) |
|--------|-------|------|---------------|----------------|
| blended | 4 | 4 | 80% | 90% |
| blended | 8 | 4 | 100% | 80% |
| blended | 8 | 8 | 80% | 80% |
| vanilla | 4 | 4 | 80% | 100% |
| vanilla | 8 | 4 | 60% | 90% |
| vanilla | 8 | 8 | 100% | 80% |

To quantify the overlap of the results, we calculate the proportion of shared attention heads in the top-5 and top-10 influential positions across LC and CC. As shown in Table 9, blended-trained models exhibit high consistency, with top-5 overlap ratios reaching 100% in some configurations (e.g. Layer 8, Head 4), and top-10 overlaps ranging from 70% to 90%. Even in vanilla-trained models, considerable overlap exists, for example: a 100% top-5 match in Layer 8, Head 8, though with slightly more variance (e.g., 60% overlap in Layer 8, Head 4). These highly overlapping results further support the idea that the model does not differentiate between individual functions but instead resolves the context in a more generalized manner.

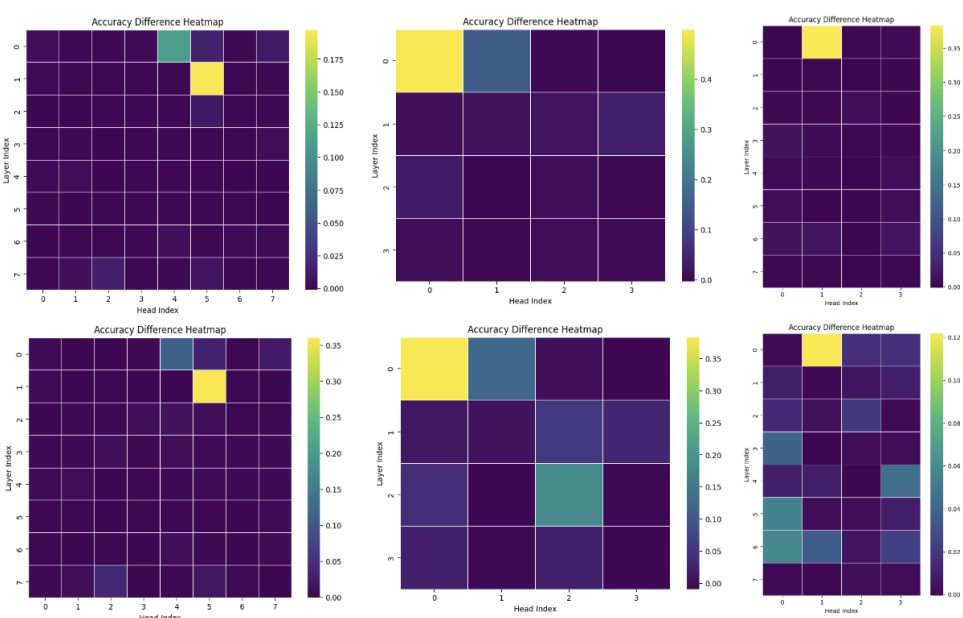

Figure 6: Vanilla-trained model accuracy difference heatmaps

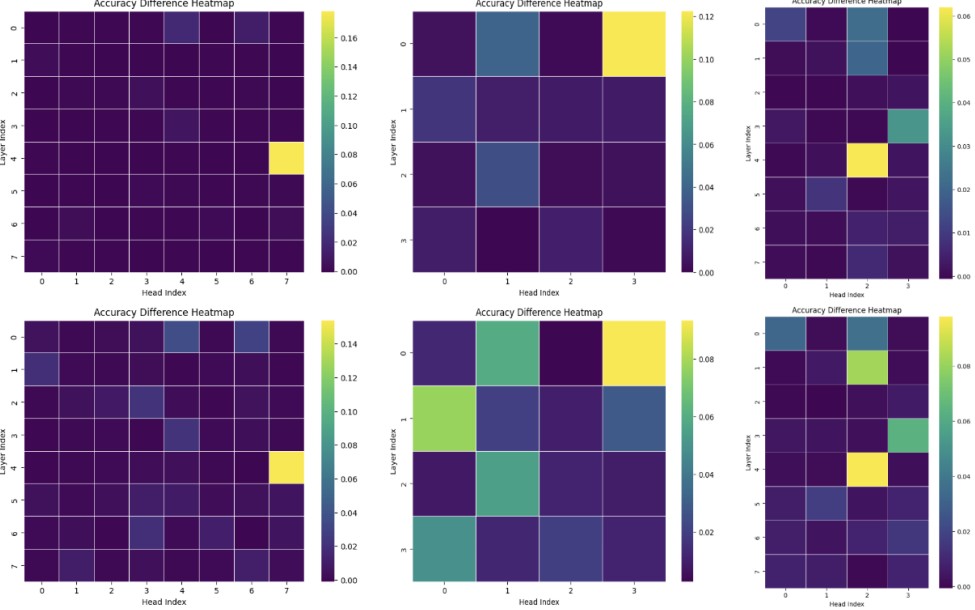

Figure 7: Blended-trained model accuracy difference heatmaps

Figure 8: More results of accuracy drops from attention head ablation across different transformer layers and heads. Left: Model with 8 heads of 8 layers, Middle: Model with 4 heads of 8 layers, Right: Model with 4 heads of 4 layers

