# OpenReview forum: "Unpacking In-Context Learning: Underlying Mechanism and Out-of-Distribution Generalization via Blended Training on Function Mixture"
_ICLR.cc/2026/Conference — Submitted to ICLR 2026_

### Official Review · Reviewer_K6Ru · 2025-10-28

**Soundness:** 2
**Presentation:** 1
**Contribution:** 2
**Rating:** 2
**Confidence:** 3

**Summary:**

This paper makes a valuable contribution to the growing literature in-context learning (ICL) in Transformers. It introduces a "blended training" paradigm for ICL. In this paradigm, models are trained on prompts containing examples sampled from multiple function classes without any explicit task identifiers, thereby creating a more structurally complex and ambiguous learning environment. It provides crucial empirical evidence that the prevalent "function selection" behavior—a cornerstone of many current theoretical explanations for ICL—becomes less dominant under blended training. Empirical work demonstrates that this blended training approach leads to tangible performance improvements, including: model robustness to input noise, stronger generalization to out-of-distribution (OOD) tasks.

**Strengths:**

1. The paper identifies and tackles a significant limitation in the current In-Context Learning (ICL) literature. This work investigates the more complex and ambiguous scenario of "blended training," which is a crucial step towards understanding ICL in more complex real-world applications.

2. The experimental design allows for a controlled and interpretable investigation of a complex phenomenon, enabling the authors to draw clear comparisons to prior work.

3. The key finding—that the dominant "function selection" mechanism diminishes in importance under blended training—is both surprising and insightful.

**Weaknesses:**

1. The presentation of this paper is poor. Many descriptions is confusing and hard to understand. The details can be found in the part of Questions.

2. The designed function type is kind of simple. I suggest to supplement more complex function class, such as MLP-based functions.

**Questions:**

1. In this paper, the word "structure" is mentioned many times. For example, "These works suggest that ICL involves structured, algorithmic behavior within the model." What's the meaning of structured behavior?

2. In line 253, the label $\hat{y}_{1,2,3...}$ are all computed. However, in the previous paragraph, only the value of y_100 is calculated.

3. In section 4.2, the evaluation protocal is vague. Why "the 100-th point was appended 2000 times"?

4. In Table 4, the meaning of each column is confusing. The author mentioned "Specifically, we compare vanilla-trained and blendedtrained models to a baseline formed by individually trained models". It seems that "baseline formed by individually trained models" are exactly vanilla-trained models.

---

### Official Review · Reviewer_ThSc · 2025-10-28

**Soundness:** 3
**Presentation:** 1
**Contribution:** 2
**Rating:** 2
**Confidence:** 4

**Summary:**

The authors investigate blended training for in-context learning, a training paradigm introduced by [1]. Specifically, they analyze how models trained with a blended training strategy generalize to OOD tasks and how they perform function selection.

[1] Yingcong Li, Xupeng Wei, Haonan Zhao, Taigao Ma. Can Mamba In-Context Learn Task Mixtures? ICML 2024 Workshop on In-Context Learning.

**Strengths:**

- The questions asked are important and relevant -- In particular, how and when models generalize to OOD ICL tasks remains a relevant question for the field
- The experiments testing the blended training setup are well-designed, and pair function classes in the blended training paradigm that are misaligned (e.g. Section 3.1)
- The authors take steps to investigate what circuits in the model are important for OOD generalization, an interesting question with far-reaching implications

**Weaknesses:**

- The paper is confusing in places, making it hard to figure out what was done:
    - The authors do not adequately describe the blended training setup. It is not clear from section 4.1 that in the blended training setup, the model (I think, from reading [1]) is also given information about which task $f_j$ is being presented to the model at the current sequence position.
- The experiments testing OOD generalization could be improved:
    - The authors choose the 'out-of-distribution' function class seemingly arbitrarily. When pretraining on some set of in-distribution functions, the out of distribution performance should depend on the interplay between the pretraining tasks and the OOD tasks, as in [2]. The authors argue that blended training improves OOD performance, but the OOD tasks considered here are too limited to make this claim.
    - It is not clear how the 'task parameters' are chosen (e.g. $w, w_1, w_2, A, \ldots$ in Table 1). Could the authors clarify this?
    - In the comparison with the noise-augmented model (Section 5.3.1), the authors pick a fixed noise level ($p=0.3$) and assert that blended training outperforms the noise-augmented model. However, it is not clear whether this is just a result of the noise level selected. Perhaps a higher/lower value of $p$ would achieve comparable performance to blended training.
- Attention head analysis: It is not clear to me from the authors' experiments that the function selection hypothesis is refuted. The authors show that there are attention heads whose presence is important to multiple function classes.
    - It is possible that the 'top-performing' heads that the authors refer to perform the function of either i) embedding or generic, task agnostic manipulation of input data to fit the model's preferred internal representation or ii) consolidation of task-specific information via some generic prediction mechanism
        - The heads the authors identify appear at early layers in the model, suggesting that i) is more likely
    - The authors' ideas are based on the observation that ablation of one head shows a large change in model accuracy, ignoring the possibility of 'delocalized' circuits


[1] Yingcong Li, Xupeng Wei, Haonan Zhao, Taigao Ma. Can Mamba In-Context Learn Task Mixtures? ICML 2024 Workshop on In-Context Learning.

[2] Chase Goddard, Lindsay M. Smith, Vudtiwat Ngampruetikorn, David J. Schwab. When can in-context learning generalize out of task distribution? ICML 2025.

**Questions:**

- How do models perform on OOD tasks as a function of how similar the OOD task is to the pretraining tasks?
- Is $p=0.3$ the best-performing choice of noise level?
- Is there a mechanistic explanation for what the heads are doing that would alleviate my concerns above?

---

### Official Review · Reviewer_M4J5 · 2025-10-31

**Soundness:** 3
**Presentation:** 3
**Contribution:** 2
**Rating:** 4
**Confidence:** 3

**Summary:**

ICL is a key capability of large language models, yet its underlying mechanism remains unclear. Prior work has primarily focused on contexts drawn from a single function class, overlooking the structural diversity and ambiguity that real-world tasks often exhibit. This paper introduces mixed training, where each training prompt contains samples from multiple function classes, without any task labels or structural cues.

**Strengths:**

- The paper proposes and systematically evaluates a mixed training paradigm, challenging the conventional single-function selection assumption. It extends ICL research from single-function settings to multi-function mixed scenarios, better aligning with real-world tasks.

- Synthetic tasks are used to control variables, facilitating mechanistic analysis. Multiple experiments are designed to test the hypotheses from different perspectives. Through attention-head ablation analysis, the paper shows that the model does not modularly select functions; instead, it shares computational resources and performs dynamic fitting.

**Weaknesses:**

- Limited task complexity: The function classes used in the study are relatively simple, which raises concerns about the generalizability of the conclusions.
- Small model scale: The experiments rely on GPT-2–sized models, limiting the universality of the findings.
- OOD tasks remain structurally aligned with training tasks: Although the paper evaluates out-of-distribution scenarios (e.g., transferring from LC/CC to R), the OOD tasks still share structural similarities with the training distribution. Fully novel task types are not explored.
- Unaddressed training efficiency: The paper does not discuss whether mixed training requires more data or longer training time.
- Readability issues in experimental results: Some results are difficult to interpret; for example, the results in Table 4 are quite confusing for the reader.

**Questions:**

- How are the points (pts) chosen in Experiment 5.2.2?
- The experiments show that attention heads are shared across tasks. Does this imply that these heads learn more fundamental primitive operations?
- Could the paper report training efficiency results? (e.g., training time, convergence behavior, or computational cost)
- How is the context length determined in the experiments? Are results available for other context lengths?
- During training, for instance in the vanilla setting, how is the ordering of mixed function-class samples within the context determined?
- How does the dimensionality of the data affect the results? What is the dimensionality of $x$ in your experimental setup? Considering that embedding dimensions in real-world scenarios are typically much higher, how does your work strengthen its claims in terms of empirical validity and generalizability under higher-dimensional settings?
- Although the experiments suggest that the “function selection” assumption does not hold, is there a new theoretical framework proposed to explain how the model achieves dynamic adaptation?

---

### Official Review · Reviewer_NRwE · 2025-11-01

**Soundness:** 2
**Presentation:** 3
**Contribution:** 1
**Rating:** 2
**Confidence:** 4

**Summary:**

This paper compares on various common ICL tasks to demonstrate that hybrid training does not degrade the model's predictive performance, in some cases it even outperforms the traditional training. It also conduct a mechanism analysis on the model's internal mechanisms under different training strategies, which reveals that hybrid-trained models are more flexible in dealing with task diversity and noise. Experiments show that hybrid training improves the model's noise robustness and OOD generalization ability, especially when training data is incomplete or tasks are diverse.

**Strengths:**

1. The paper breaks through the limitations of traditional ICL research, which only trains and evaluates on a single function or task distribution.

2. It adapts the concept of Blended Training, where each training cue consists of a mixture of samples from multiple function categories, without explicit task labels. This setting more closely resembles the real-world scenarios of ambiguous and structurally diverse tasks, thus possessing strong practical significance and theoretical values.

**Weaknesses:**

The paper almost does not propose any method, the proposed masking-based diagnostic is too naive. It indeed has a mechanism analysis through controlled experiments, but there's no any theoretical analysis about it, and I don't think this mechanism analysis alone is enough for a conference like ICLR.

**Questions:**

None

---

### Official Review · Reviewer_Z35Y · 2025-11-04

**Soundness:** 1
**Presentation:** 1
**Contribution:** 2
**Rating:** 2
**Confidence:** 3

**Summary:**

This paper studies in-context learning (ICL) under a new blended training paradigm where each prompt mixes examples from multiple function classes without task identifiers.

The goal is to test whether models trained this way can go beyond “function selection”—the hypothesis that Transformers identify and apply one underlying function per context—and instead demonstrate more flexible adaptation and better out-of-distribution (OOD) generalization.

Using small GPT-2–style Transformers on synthetic datasets (linear, quadratic, checkerboard, and residual classification), the authors compare vanilla (single-function prompts) vs blended (multi-function prompts) training.

They report that blended training:
1. matches vanilla performance on seen tasks,
2. generalizes better to unseen function mixtures,
3. exhibits more robust behavior under input noise,
4. and uses overlapping attention heads across tasks—suggesting shared, rather than function-specific, representations.

**Strengths:**

1. The paper targets an important and underexplored question: how ICL mechanisms change when the context combines heterogeneous task structures.
2. Multiple diagnostics are considered—accuracy, OOD generalization, attention ablation, and noise robustness—offering a multi-angle empirical view.

**Weaknesses:**

1. The paper is difficult to read and often imprecise. Key issues include: Long, unfocused paragraphs that mix motivation, results, and speculation. Undefined or vague terms (e.g., “lowest-error preference” in "Mechanism analysis" contribution). Ambiguous phrasing like “the model demonstrates more flexible pattern recognition” without quantitative evidence. Overall, the text reads like a draft in progress, not a polished ICLR paper.

2. The Related Work section misses major lines on ICL mechanism as function learning such as: Function vectors (Todd et al., 2024; Hu et al., 2025), In-context gradient descent (von Oswald et al., 2023; Ahn et al., 2023).

3. In table 2-3, no baseline for training on explicit mixtures or multi-task pretraining is provided, making it unclear whether improvements come from blending or merely data diversity.

**Questions:**

1. What does this paper add beyond Li et al. (2024b)? Is it a new mechanism, a stronger empirical validation, or a diagnostic framework?

2. Each subsection introduces H1/H2 vaguely. Can you provide mathematical or schematic definitions of what “function selection” vs “context adaptation” means?

---

### Meta-Review · Area_Chair_3ieG · 2026-01-07

**Summary:**

This paper investigates the mechanisms of In-Context Learning (ICL) in Transformers through a "blended training" paradigm, where prompts consist of samples from mixed function classes. The authors argue that this approach moves the model away from a "single-function selection" mechanism toward a more flexible pattern recognition strategy, leading to improved Out-of-Distribution (OOD) generalization and noise robustness.

**Reviewer Concerns:**

The reviewers consistently criticized the paper for poor presentation, vague terminology, and a lack of technical depth. Major concerns included the simplicity of the synthetic tasks, the absence of theoretical analysis, and the failure to provide adequate baselines to distinguish the effects of "blending" from simple data diversity.

**Reviewer Scores:**

The consensus was negative, with scores primarily at 2 (Reject) and one 4 (Marginal Reject).

---

### Decision · Program_Chairs · 2026-01-26

Reject